# The Confluence Between Spiritual and Mental Health: A Phenomenological Approach to the Study of Healthcare Professionals’ Experiences

**DOI:** 10.3390/healthcare13010035

**Published:** 2024-12-28

**Authors:** Ángeles C. López-Tarrida, Paola Suárez-Reina, Rocío de Diego-Cordero

**Affiliations:** 1Department of Critical Care and Emergency, Hospital Saint John of God Aljarafe, Hospitaller Order of Saint John of God, Bormujos, 41930 Seville, Spain; 2Virgen de Valme University Hospital, 41014 Seville, Spain; 3Faculty of Nursing, Physiotherapy and Podiatry, University of Seville, 41009 Seville, Spain; rdediego2@us.es

**Keywords:** mental health, spiritual care, spiritual needs, spirituality, religiosity, healthcare professionals, mental illness

## Abstract

**Background**: Given the global concern about mental health in the world, different approaches are being explored in its approach and treatment. In this line, the care of the spiritual dimension has been shown in many studies to have a significantly positive relationship. In mental health units, the comprehensive approach that involves comprehensive care considers the spiritual dimension as an aspect of care that contributes to coping with mental health problems. **Methods**: This is qualitative research with a descriptive design and an ethnographic approach, using interviews with forty-five professionals from Spanish and Portuguese mental health units. **Results**: The professionals do not define the term spiritual health in the same way; however, all interviewees believe that S/R positively influences the mental health of their patients, although few address it. They believe that their own S/R can influence their attention to spiritual needs. Among the limitations are the lack of training and time due to the prevailing biomedical model. Lack of time and specific training in spiritual care are the main aspects to which they attribute the shortage in meeting spiritual needs. Most of them expressed feeling challenged to care for the spiritual dimension after this research. **Conclusions**: More studies are needed on the spiritual care provided by mental health professionals to specify specific training and the associated challenges in this field.

## 1. Introduction

The inexhaustible debate surrounding the concept of mental health extends throughout history in a continuous attempt to achieve a definition that encompasses its complexity. Just as the term health has evolved and changed over time, mental health is not only part of this well-being (biopsychosocial and spiritual balance) but is understood as an independent concept.

The state of health, understood from a holistic perspective, recognizes that ‘the human being is more than the parts of the self’, being the result of a dynamic balance of these parts that include body, mind, spirit, and environment. Divisions or imbalances within this human structure are what produce tension or illness [1].

Nowadays, and returning to the above, mental health is considered an integral and vital component of the emotional, psychological, spiritual, and social well-being of human beings [2]. Moreover, it refers to the way people behave, identify themselves, and the way they experience and cope with life events [3].

According to the World Health Organization (WHO), 1 billion people suffer from mental disorders worldwide (Mental Health: Strengthening our response (who.int), accessed November 2024), making the perception of such issues more evident, especially with the 25% increase in incidence since the COVID-19 pandemic.

Spirituality, a quality inherent to the human being, is part of the person in all areas of life and at any vital moment through which he or she is passing. This is another concept that is difficult to categorize according to the context, even more so if it is addressed in healthcare in clinical settings, as shown by the existing literature, in which various authors from different perspectives have discussed the subject, since there is great confusion regarding the concept, as there is no clear distinction between spirituality, religiosity, and, in some research, with well-being [4,5,6,7,8,9,10,11,12,13,14,15]. In mental health, this dimension seems to be closely related to the sense and meaning that is given to what happens to us and must be considered as in any other clinical field; there is research that indicates and demonstrates that spirituality also plays a relevant role in people with mental disorders [16,17,18,19]. Spiritual health has been found to lead to better mental health and to positively influence physical health [20], and some studies have even shown the benefits of spirituality/religiosity in mental health, suggesting that it is especially effective in lowering levels of anxiety, stress, and depression [19,20].

Although research in the field of spirituality and health has grown substantially in recent decades, there are still few health professionals who address this issue and include it in their clinical practice [21,22]. There is research that shows the limitations and barriers that professionals encounter when addressing the spiritual dimension in their healthcare practice, and even considering its importance, the main shortcoming is the lack of specific training [6,21,22,23,24]. In a specific study conducted among mental health professionals [23], it is concluded that, even though there is no agreement on the definition of spirituality, professionals believe that attention to this human dimension has a positive influence on patients’ health when facing illness. The professionals consulted think that their own spiritual dimension may interfere in the clinical relationship with spiritual needs. They said that lack of time and lack of specific training in spiritual care are the main reasons why few of them take this facet into consideration, and if these obstacles were overcome, spiritual care in the mental health field would improve. Other research that addresses the influence of spiritual care in mental health reflects the positive impact of addressing this dimension [20,25].

In clinical settings in Spain and Portugal, as in other European countries, comprehensive care of the person is understood as a construct that considers physical, psychological, social, and spiritual aspects. However, the way health systems are set up is to ensure care is diverse, even more so if they are public or private entities. In the public health systems of Spain and Portugal, for the care of the spiritual and/or religious dimension where the sociocultural context is similar, there are chaplaincies that support this care, aided by priests related to the Church, in some cases, or by lay people with specific training in spiritual care with their own professional status (counselors).

In the centers of the Hospitaller Order of St. John of God (HOSJD) in particular, where this research has been carried out, there are specialized departments called Spiritual and Religious Care Services, which are mostly run by counselors who collaborate with priests to ensure both spiritual and religious care.

Continuing the research started in 2022 [23] on the perspective that mental health professionals have on the approach and impact that mental healthcare has on their patients, this study was conducted by expanding the sample and analyzing in depth by categories the answers and the data provided.

The goal of this study was to explore the views, opinions, and behaviors of mental health professionals in Spain and Portugal regarding the care and attention to the spiritual needs of patients and their families in the mental health field, and with this data, we can implement future improvements.

## 2. Materials and Methods

A qualitative, exploratory, observational study has been conducted using a phenomenological approach, which aims to describe the meaning of an experience by identifying themes and sub-themes born out of participant observation and discourse [26,27].

Data collection was conducted through in-depth interviews conducted by three researchers for 8 months at two separate times: November 2021 to February 2022 and from January to February 2023, due to the availability of interviewers and participants. This study was registered in OSF (Identifier: https://doi.org/10.17605/OSF.IO/HQN4P, accessed on 27 December 2024).

The profile of the participants was healthcare professionals from the mental health services and centers of the Hospitaller Order of Saint John of God (HOSJD) in different centers in Spain and Portugal. Convenience sampling was conducted through key informants who allowed the participants to be approached, pursuing the inclusion of different profiles in terms of gender, age, length of experience in that unit, etc.

Participants were invited to participate in this study on a voluntary basis and were included in the study after signing the free and informed consent form.

A script was used to conduct interviews, drawn up by a group of experts in spiritual health and mental health, who carried out a prior analysis of the content of the script to assess its suitability and made the necessary adaptations.

Applying a phenomenological-hermeneutic paradigm (following Amadeo Giorgi’s theory [28]), the qualitative approach was carried out through the realization of interviews, following this previously agreed-upon script. Regarding the thematic analysis, a partly inductive and deductive approach was carried out; the previous script was used for the interviews with predetermined variables, which were combined with themes and subthemes that arise from the observation and the discourse of the participants in the interviews.

The interviews were recorded and transcribed. Following this described analysis, the following categories were identified: P-xx (code for each participant to keep confidentiality), sex, age, and professional category. The qualitative analysis was realized following the steps proposed by Braun et al.: (a) familiarization with the data; (b) generation of categories; (c) search, review, and definition of themes; and (d) final report [29].

This research followed the criteria of the Consolidated Criteria for Reporting Qualitative Studies (COREQ) [30].

In addition, after the data classification and analysis phase, a triangulation of the identified categories was conducted by the members of the research team.

Appropriate ethical considerations were considered, and this study was approved by the Research Ethics Committee of Andalusia, Spain (Code: 0731-N-19). The participants who were invited to take part by the researchers signed the relevant informed consent once they were properly informed of the aims and the development plan of the project. In turn, all possible doubts about it were resolved, both by telephone and through a Participant Information sheet. In addition, this research follows the Declaration of Helsinki 1964 and its later modifications or comparable ethical standards. Furthermore, all information collected is protected by Organic Law 3/2018, of 5 December, on the Protection of Personal Data and Guarantee of Digital Rights (BOE-A-2018-16673 Ley Orgánica 3/2018, de 5 de diciembre, de Protección de Datos Personales y garantía de los derechos digitales).

## 3. Results

A total of 45 participants made up the final sample, all of them professionals in the field of mental health in the HOSJD in Spain and Portugal, being active at the time of the interview.

Of the total, 30 were women and 15 men. In terms of origin, 5 were from Portugal and 40 from Spain. The mean age of participants was 46.6 years (age range 23 to 61 years, DS 10.6). The mean number of years of professional experience in mental health was 20.8 years (time range 3 to 38 years, DS 10.7).

About the professional category, there were 13 nurses, 8 psychiatrists, 12 psychologists, 4 nursing assistants, 2 social workers, 4 occupational therapists, and 2 counselors.

All of them had a postgraduate degree or a specialized degree in the field of mental health and knowledge of spirituality obtained within the HOSJD. Regarding their own spiritual and religious beliefs, 100% defined themselves as Catholic.

Initially, some categories were planned, which were expanded with other emerging categories that appeared during the interviews and are presented below along with the most relevant results.

### 3.1. Spiritual Health: What Is Known

There is no agreed definition among the participants, and most of them (except two) had not heard of the term itself. However, when questioned about the meaning they attribute to it, different perspectives can be established, with various aspects prevailing.

On the one hand, it is related to well-being and quality of life, in line with the state in which one feels good about oneself:


*“it is something beyond mental well-being, it almost gives you a certain tranquility and a certain relaxation of what may be the overall concept of life”*
(P-5, man, 47 years old, nurse)


*“I understand that it is a state of inner wellbeing, of balance… we have many concerns, many questions, much inner anxiety and it is like being at peace with yourself.”*
(P-12, woman, 54 years old, psychiatrist)


*“to find oneself and develop in such a way that body and mind are aligned”*
(P-10, woman, 29 years old, nursing assistant)


*“it is the sense of quality of life, quality of emotional well-being and being with myself in the world”*
(P-23, man, 48 years old, nurse)

From another perspective, spiritual health involves a relationship with values and beliefs. Most of the participants proved this with their own moral and/or ethical code of conduct:

“*being clear about one’s values”*
(P-9, woman, 60 years old, psychiatrist)

“*to be coherent with one’s values, something that has to do with one’s ethics”*
(P-19, woman, 47 years old, psychologist)

“*that the person is aligned with their principles… that their life is aligned in this sense”*
(P-32, man, 51 years old, psychologist)

On the relationship with transcendent meaning and significance, participants showed the following:


*“as another need of the person, when they come here to be admitted, just as there is a physiological need for psychological health, there is also the need for transcendence”*
(P-22, man, 50 years old, nurse)


*“to answer deep questions of identity, and of vital projection”*
(P-6, woman, 46 years old, psychologist)


*“to be clear, where you are going and to have something that gives meaning to your life and allows you in moments of difficulty throughout your life”*
(P-9, woman, 60 years old, psychiatrist)


*“… if it makes sense to be here and why”*
(P-11, woman, 56 years old, psychologist)


*“what are we doing here, where are we going, where do we come from, what is the object of our life, or what things concern us, fill us, satisfy us, worry us in our life”*
(P-24, man, 62 years old, psychologist)

Finally, they related it to a feeling of connectedness:


*“the deep capacity for connection with oneself, and contact with the other”*
(P-6, woman, 46 years old, psychologist)


*“the connection with oneself and with the whole, understanding the whole as nature, people and something higher that includes and embraces us all”*
(P-8, woman, 57 years old, nurse)


*“clean and open connection with something bigger than ourselves, with a whole”*
(P-17, woman, 60 years old, psychologist)

### 3.2. Influence of Spirituality in Clinical Practice: Barriers and Facilitators

Regarding the lack of time attributed to a lack of integration of spiritual care in clinical protocols, participants stated the following:


*“the rhythm of the day takes up everything… the things that have to be done in protocol take time away from other things more for the person”*
(P-19, woman 47 years old psychologist)


*“Sometimes there is not enough space or time”*
(P-6, man, 47 years old, psychologist)


*“it is not a practice that is totally integrated in the care”*
(P-37, woman, 59 years old, psychiatrist)


*“for me the main barrier is time”*
(P-14, man, 57 years old, psychiatrist)

They showed the absence of training in spiritual care, which makes it difficult to identify the spiritual dimension:

“*there are times when it is not easy in mental health”*
(referring to identifying the spiritual dimension) (P-30, woman, 42 years old, psychologist)

“*there is a lack of training in distinguishing concepts such as spirituality and religion”*
(P-36, woman, 29 years old, nursing assistant)

“*when presented with a problem of this kind, look the other way… And this may be present in some professionals in the sector because there are issues that perhaps make us uncomfortable, and we prefer to say well… this for another professional in another field and I’ll forget about it”*
(P-22, man, 50 years old, nurse)

“*lack of knowledge and not having the tools to be able to deal with it”*
(P-26, woman, 46 years old, nurse)

“*I have doubts due to lack of training”*
(P-15, woman, 31 years old, nurse)

“*the main barrier is my knowledge of the subject, you have to train yourself to be able to work on it well”*
(P-2, man 30 years old, nurse)

“*we need better training”*
(P-12, woman, 54 years old, psychiatrist)

The consideration that spirituality is not related to clinical practice was pointed out as a barrier, either because it is seen as a vetoed subject or because it is something difficult to quantify:


*“it is a dimension that has a lot of taboo…’. I may not be aware of it because it is intangible, I overlook it even though I am sensitive”*
(P-9, woman, 60 years old, psychiatrist)


*“stereotypes and prejudices… everyone is moved by what they have lived through”*
(P-16, woman, 39 years old, counselor)


*“it is a subject that is kind of behind and you talk about it in an indirect way”*
(P-21, woman, 43 years old, occupational therapist)


*“it is an abstract, delicate dimension”*
(P-19, woman, 47 years old, psychologist)

It was also identified as a barrier that the person being addressed does not request or refuses attention to their spiritual dimension:


*“resistance from the patient because he is not prepared”*
(P-5, man, 47 years old, nurse)


*“the patient can say yes or no”*
(P-11, woman, 56 years old, psychologist)


*“the person does not accept the referral because they say that they do not need to be treated in this dimension”*
(P-12, woman, 54 years old, psychiatrist)


*“the most complicated thing is that the person does not identify his spiritual dimension”*
(P-32, man, 51 years old, psychologist)


*“it is not easy…, we are held back by the fear of encountering the anguish of the other”*
(P-19, woman, 47 years old, psychologist)

As facilitators for spiritual care, it was pointed out that mental health could be an ideal place to work on spirituality: “*we don’t have to pay so much attention to the physical, as happens in other realities… we have that privilege”* (P-43, man, 40 years old, nurse).

### 3.3. Spiritual Care in the Mental Health Field

They highlighted the importance of spirituality in the management of mental health problems, arguing that it can have an influence when it is understood or related to a person’s life crisis:


*“when a person’s world or understanding of the world around them is broken, it affects in one way or another where I am going and what meaning my life has with what is happening to me… the person who has a more integral spiritual health structure, sometimes that gives them strength, even if at that moment the mental health problem makes them question everything, which is when they ask you for help”*
(P-9, woman, 60 years old, psychologist)


*“it is useful for those people who at some points are in crisis, do not see the meaning of their life and want to die… when there is suffering”*
(P-12, woman, 54 years old, psychiatrist)


*“a mental health crisis ends up misaligning everything, there are even times when the decompensation comes from a misalignment of these aspects… And a mental health crisis is a life crisis in terms of values, beliefs), or according to the meaning that the mental illness has for the person”*
(P-39, man, 34 years old, psychiatrist)


*“it helps him to understand a condition that may be caused by the illness, to find the meaning of what is happening to him and to help him, as it is a stigmatized and rejected illness and having this part well worked on would help him”*
(P-23, man, 48 years old, nurse)


*“it often helps to clarify aspects of their human condition, worries and problems”*
(P-24, man, 62 years old, psychologist)

They also think that it can be a source of help and reassurance for people with mental health problems:


*“mental illness generates a lot of rejection and feelings of loneliness and the connection with something that is above me and from which I feel cared for can be beneficial”*
(P-6, woman, 45 years old, psychologist)


*“and once they connect with something and are able to see themselves and others, they can realize that they are part of something, that they are not alone”*
(P-8, woman, 57 years old, nurse)


*“it is also important not to stigmatize, not to label that there is something beyond the disease, the symptom or the diagnosis) and as a therapeutic tool for treatment”*
(P-27, woman, 47 years old, nurse)


*“having more capacity for introspection and a better version of oneself, being able to solve social, family and relationship problems, everything that helps you to generate inner peace will help the waves of life catch you stronger”*
(P-6, woman, 45 years old, psychologist)


*“it is essential, for some it is even the only way out”*
(P-17, woman, 61 years old, psychologist)

On the influence of one’s spirituality on clinical practice, they recognize that practitioners who have a more nurtured spirituality provide better spiritual care:


*“you need to see yourself to see. What you don’t work on yourself you can’t see in others. In fact, it can be trained”*
(P-8, woman, 57 years old, nurse)


*“if I don’t know the term spiritual health, it is clear that I won’t know how to help them”*
(P-15, woman, 31 years old, nurse)


*“the spirituality of the health professional influences the therapeutic relationship, I am convinced, the more spiritual health professionals provide different care… I believe that a more spiritual person will be more receptive, let’s say to the spiritual demands of the patient. And even if there is no time, it will be better dealt with”*
(P-29, man, 52 years old, psychiatrist)

It is also pointed out that strengthening spirituality itself makes professionals more sensitive to this care:


*“in situations where you perceive it you try to address it obviously… but we who have people admitted for a more or less long time if we see that this situation is present and we are sensitive to them because well I think that factor is also the sensitivity a bit of the professional who is being trained when faced with these circumstances”*
(P-29, man, 52 years old, psychiatrist)


*“you are more receptive to demands in this area if your experience is familiar. Another person who does not have this feeling, normally overlooks or goes to other people who are more in tune with this perception… More than worked, the one who has been educated and has this belief, is more sensitive to this type of care, there are people who avoid it…”*
(P-33, man, 45 years old, nurse)

They admitted to including this type of care in their clinical practice:


*“they come to us and come with high levels of anxiety, and a closer treatment, contact, talking to them, listening to them, trying to get them to express their feelings… we do relaxation exercises, and they start to calm down”*
(P-10, woman, 29 years old, nursing assistant)


*“I explore according to the convictions and their intensity. I integrate what I deduce or what he/she has explained to me, if it is useful”), although there are occasions when it is not treated directly”*
(P-37, woman, 59 years old, psychiatrist)


*“I avoid it if I see them totally reluctant, or because talking about it would impede the relationship or that it would not be beneficial for the bond in some way”*
(P-32, man, 51 years old, psychiatrist)


*“I don’t actively avoid it, but I do redirect it, use it in a way that doesn’t enhance the pathological part”*
(P-37, woman, 59 years old, psychiatrist)


*“we have a spiritual assistant who uses the FACIT scale to deal with this issue”*
(P-42, woman, 42 years old, psychologist)


*“using scales (FACIT), putting this on paper, helps to work on these issues”*
(P-40, woman, 40 years old, nurse)

### 3.4. Spiritual Health: The Unfinished Business

Despite the fact that the professionals interviewed have no experience or training in spirituality, they agree that further strengthening this aspect would exponentially increase the quality of treatment in this area:


*“yes there should be more, for example, in workshops prior to certain subjects related to basic care”*
(P-2, man, 30 years old, nurse)


*“training should be a pillar. It is not rewarded, it is punished. Although if it is received as an imposition it can be counterproductive”*
(P-8, woman, 57 years old, nurse)


*“through courses or seminars, or before from education, so that they (talking about the future professionals) have this skill from an earlier age”*
(P-10, woman, 29 years old, nursing assistant)


*“it is something that cannot just be done and close. It should be done on a more continuous basis, perhaps when you start working and have more contact with the patient, rather than during your career”*
(P-19, woman, 47 years old, psychologist)

Some consider that it is important to clarify the terminological confusion between religion and spirituality for a better understanding of this issue:


*“spirituality is often associated with abuses of power by the Church”*
(P-17, woman, 61 years old, psychologist)


*“I would do more training, but perhaps by changing the name, as many associate it with religion and that creates rejection”*
(P-27, woman 47 years old, nurse)


*“not all professionals have the same capacity and giving training to everyone and saying that everyone does it because it is linked to the capacity of the professional sounds false”*
(P-31, woman, 54 years old, psychiatrist)

In this regard, participants highlighted as the main measure for spiritual care the training of professionals:


*“train professionals so that they can explore and help patients to develop their spiritual dimension”*
(P-38, woman, 45 years old, psychologist)


*“for me, what would be fundamental to reach them better: more and better training”*
(P-16, woman, 39 years old, counselor)

Together with this, the importance of research studies in relation to spiritual care in the clinic, specifically in mental health and that of personal and professional interpellation in relation to the spiritual dimension, is emphasised:


*“I appreciate the research on spirituality that links training, it is that it is basic to improve information and documentation in this regard”*
(P-16, woman, 39 years old, counsellor)


*“the mere fact of having done this interview is good for me because it makes me ask myself questions”*
(P-9, woman, 60 years old, psychiatrist)


*“these were questions that made me think”*
(P-30, woman, 41 years old, psychologist)

## 4. Discussion

The target of this study was to investigate the views, opinions and behaviors of HOSJD mental health professionals in Spain and Portugal in relation to the care and attention to the spiritual needs of patients and their families in their clinical practice. HOSJD centers around the world have the same comprehensive care model that considers the spiritual dimension as part of restoring health. The results show a lack of consensus among participants on the definition of spiritual health, presenting diverse perspectives related to quality of life, values, meaning of life, and sense of connectedness. However, all participants, from their own experiences, believe that addressing the spiritual dimension positively influences coping with mental health problems.

A relevant fact is that all the professionals who took part in the study defined themselves as religious and Catholic, which makes us consider that those people who are believers reveal a more accentuated sensitivity to the care of the spiritual dimension in clinical practice. This supports what other studies have reported on this subject [31,32]. It is also clear that women are more sensitive to this issue [22,23], although this may be since there is a larger population of women than men in the healthcare field, and our sample is proof of this.

The professionals interviewed showed that there is a significant correlation between spiritual care and coping with illness, leading to an improvement in quality of life. This finding is echoed in other research in other healthcare settings, such as the study by Camargos et al. [33] in oncology, where in matched samples of patients and professionals, both agree on the benefit of spiritual care in their healthcare. Regarding quality of life, it was observed that patients receiving treatment that included a spiritual approach had better health outcomes. A case-control study comparing the effects of spiritual interventions in cancer patients [34] found a statistically significant difference in outcomes for those in the intervention group. Also, in a mental health study on depression [35], people who used religion as a coping mechanism for their illness had a lower risk of suicidal ideation and better control of their mental disorders.

The data obtained in our research indicate that the way in which the professional takes care of his or her spiritual dimension influences the willingness to attend to the spiritual needs of patients; professionals who are more sensitive to these issues are more likely to attend to them. As evidenced in other studies [6,36,37,38], those professionals who have a more nurtured spirituality are more sensitive to caring for this dimension, which in turn fosters a better bond in the clinical relationship and more opportunities for dialogue. In the study by Neathery et al. [39] involving mental health nurses, they themselves report their own spirituality/religiosity as significant factors in providing spiritual care. In this sample it was noted that it was those who identified themselves as ‘spiritual and religious’ provided spiritual care more frequently than those who identified themselves as ‘spiritual but not religious’.

On this point, the results we found show that the majority of professionals support attention to the spiritual dimension in their day-to-day work, especially if they experience spiritual suffering, although few do so at present. There is a growing body of research in the field of spirituality and its care that supports improvement in the clinical approach, but there is still limited attention paid to the specific training of professionals, probably due to the biomedical model rooted in our healthcare system [40]. Therefore, there is a lack of spiritual care, even though most patients express a desire to discuss these issues in their relationship with health professionals.

Among the main barriers found in our study to addressing the spiritual dimension were lack of time, insufficient training, the feeling that it is a private matter, difficulty in identifying spiritual needs, patient rejection, and fear of losing control of the situation. In our context, about barriers to providing spiritual care in mental health, almost one third of the participants in our study reported experiencing challenges that deter them from addressing the spiritual dimension with their patients. In general, participants mentioned lack of time as one of the main constraints to aiding spiritual needs. This is consistent with other studies [41,42,43,44,45,46,47]. In particular, Chen et al.’s [48] study of nurses showed lack of time as the main barrier, with most of them writing down that physical medical care is more important than psychosocial care, and that spiritual needs are only addressed if there is ‘extra time’.

In the study by de Diego et al. [49] of nurses in intensive care units and emergency departments in Spain during the COVID-19 pandemic, they noted that spiritual care is provided by nurses. The nurses considered it an essential aspect of helping patients in these units but did not feel adequately prepared to provide suitable care in crisis situations. Like our findings, nurses point to lack of time and specific training as barriers [6,22,23,50].

Regarding other limitations identified in our research, a study with social workers also showed a concern about having conversations about religious and/or spiritual interests with their patients for fear of offending. This aspect made them feel uncomfortable, as it was considered a taboo subject or an intimate aspect of the person. [50].

When participants in our study were asked about how to improve or remove these barriers or difficulties, they suggested specific training measures and proposed a shift from the traditional biomedical model to a holistic model involving the psychological, social, and spiritual dimensions of the person, with an increase in resources to do so.

A relevant fact pointed out by the participants is the need for adequate specific training in the spiritual approach to clinical practice, as they consider it a clear limitation to perform it in a professional way in mental health units. As observed in another research on the subject, this is the main obstacle reported, regardless of the clinical context [10,22,23,24,34,51,52,53].

The incorporation of specific training courses in addressing spiritual needs helps professionals to be better prepared when providing spiritual care. This improves the assessment and holistic care of patients, and several studies have shown this to be the case. One of them, conducted on Spanish students, showed that they felt ill-prepared, as they claimed that universities were not providing enough training in this aspect of clinical practice. Nursing students were most likely to believe that spirituality had a positive influence on health and, therefore, the appropriateness of including it in clinical practice [6]. Training in spiritual care in clinical practice generates relevant changes in nurses’ behaviors, improving the assessment and monitoring of spiritual needs, along with referrals and care by other professionals [10,51]. It also provides essential tools to address the spiritual in relation to patients and themselves and has been demonstrated previously [34,51,52,53], showing that it has a positive impact on holistic and humanized care [54,55].

Academic training that includes spiritual care in a regulated and appropriate way is essential, as it has a positive influence on the relationship between professional and patient, generating an atmosphere of trust in clinical decision-making [32]. There are several initiatives that reveal the interest generated to improve this fact [34,51,52,53]. One of the key issues would be to clearly define the term spirituality in clinical contexts, since, as has been mentioned, having clarity in the concept favors more adequate training [4,5,6,7,8,9,10,11,12,13,14,15].

The review of participants’ statements clearly indicates that the presence of counselors in healthcare settings positively influences mental health professionals considering the spiritual dimension in their daily practice. The presence of trained and qualified professionals in spiritual care promotes a continuous awareness of these issues, highlighting the intangible aspects of human existence that are also affected by illness. This dimension can represent both a space of need and a source of coping resources, which cannot be overlooked, even in a clinical context. Specifically in mental health, the loss of meaning and purpose, the breakdown of relationships, and the disintegration of personal values impact individuals’ sense of well-being and quality of life, leading to increased distress. Skilled support is therefore essential to facilitate coping with the suffering that arises in these circumstances.

As the participants specify themselves, it is necessary to continue conducting studies that explore the spiritual dimension in clinical settings, particularly in the field of mental health [23]. In our specific study, the statements from the professionals underscore the importance of raising awareness in the care of this health area and highlight the significance of research that brings visibility to this reality from the perspective of the care providers themselves, using an interdisciplinary approach.

## 5. Conclusions

According to our study, professionals believe that it is essential to consider the spiritual in clinical care in mental health units, although lack of time, lack of training, and the biomedical model that inclines care towards the physical are the main limitations they find to this.

The absence of adequate training in spiritual care has been identified as an important predictor for caregiving, although most participants had received some type of training aimed at spiritual care within the HOSJD. We believe that more research in this area is needed to better define this, like other research that reviews this topic [56,57,58]. Qualitative research would provide further information to identify challenges and shape practical training in spiritual care needs, especially in regards to what is spirituality within the clinical environment, to improve in their care.

Finally, the professionals interviewed themselves have expressed that the experience they had with this research has encouraged them as a personal and professional challenge regarding the care of the spiritual dimension in clinical practice, and a certain awareness has increased in them in this regard.

## Data Availability

The data presented in this study are available on request from the corresponding author.

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
