# Peer review of "The Confluence Between Spiritual and Mental Health: A Phenomenological Approach to the Study of Healthcare Professionals’ Experiences"

_healthcare, 2024, doi:10.3390/healthcare13010035_

Round 1
Reviewer 1 Report
Comments and Suggestions for Authors
Some suggestions for improvement:
Line 126. I think that it is not true that all the interviewees have a postgraduate degree in Mental Health, since not all the interviewees have a university degree (nursing assistants, occupational therapists, etc.). It would be more accurate to indicate that they have a postgraduate degree or a specialized degree in Mental Health.
Line 128. All the interviewees define themselves as Catholics and also work in a Catholic center. I think that this fact should be noted and analyzed at some point in the research since it may be a bias for the results or give meaning to the study itself.
Line 517. Correct the typo in reference 10.
Author Response
Thank you for your contributions and comments to improve our article.
- You are right about the specification of the training of the professionals who participated in the study, we proceed to detail what you have referred to (now line 150).
- Regarding what you have referred to regarding the bias related to the fact that they are Catholic participants, it has been included as requested (lines 377-384) adding references in this regard.
- As you have indicated, the typographical error in the cited reference has been amended (now reference 20).
Reviewer 2 Report
Comments and Suggestions for Authors
I think the authors succeeded in presenting what expert practitioners do with the murky concept of "spirituality" and the confused and vague ideas about "spiritual care." I am familiar with the American literature, and it is refreshing to learn that Spanish and Portuguese fare no better in producing a specific enough understanding of these terms. They want more training: but how do you "train" when the real content of training is so all-over-the-map? Their research subjects are quite articulate in some ways, but the concept itself is, I am sure, too broad and ill-defined to be "trained" for.
An interesting point in this study: all their subjects were Roman Catholic, but not one of the quotes from them invokes explicitly Catholic practices or content. They do not use examples from the rich traditions of prayer and meditation, but are resolutely secular in all their statements. Use a more tradition-oriented definiton of "spirituality" and it might be possible to train someone for it (which is what "spiritual direction" is about). A trainee and a guide could then work toward some specific goal, and mark progress or setbacks. But the current way that "spirituality" is used, it cannot fit into that model of activity.
I would actually prefer that these authors substitute the word "existential" for "spiritual," and that would help guide them to systems of psychology which depend on terms such as "meaning" and "connection." I think some North European scholars use "existential" where Americans use "spiritual," because it avoids confusion over how "spiritual" and "religious" relate. This is a suggestion that could be mentioned in the paper, or perhaps best left for a further inquiry by these authors.
Author Response
We deeply appreciate your comments to improve our article, and we respond to each one of them.
- As you mention and provide examples, in the review of the existing literature regarding the concept of spirituality used in the field of health, we found multiple meanings and confusions, especially with the concepts of religion and the existential. We have added it with a brief comment in the introduction of the article (lines 52-58) since we are aware of this complexity although it is not the purpose of our research.
- We objectively present the results obtained in our sample, in the environment of Spain and Portugal, subject to the responses of this group of participants, in relation to the concept of spiritual health that they have and their attention in practice (if they attend to it, if they consider it relevant...). We have not focused on what the practices that professionals carry out to care for the spiritual dimension of their patients consist of, an aspect that you refer to very well but that we believe is interesting to address in the future.
- We find your comments on how to train or educate professionals in spiritual care very interesting if this concept is not well understood. Following your suggestion, we have made a review in lines 464-466, although this is something that has been the goal of other previous research and probably of others to be carried out, which encourages us to continue along this line.
Reviewer 3 Report
Comments and Suggestions for Authors
Dear Authors,
Thank you very much for your article, in which you examined the views and attitudes of health care providers in Spain and Portugal (working in mental health care) toward spirituality in their work. Indeed, based on research, spirituality appears to be an important factor in the experience and coping of mental health symptoms and disorders. Although it is not always clear to what extent spirituality and mental health are two separate constructs, and although the results of research are not always uniform. If little is yet known about this role in a specific context, it is indeed good to conduct initial explorations through qualitative research. By and large, the article runs well, but I still have a number of larger and smaller questions and comments that I hope you can/want to update the article with.
1. It seems to me that the introduction (and perhaps also the title) should emphasize more strongly the fact that the research took place in Spain (and to a lesser extent in Portugal). It also seems to me that that context could be fleshed out a bit more. What is the state of affairs in Spain/Portugal when it comes to the appreciation of the role of spirituality in the treatment of mental illness? To what extent is there in Spain/Portugal the profession of spiritual caregivers or chaplains who can play a role as experts in or parallel to treatment? In this case, I understand spiritual care to mean a profession relatively separate from the churches, which has its own professional status, and whose representatives are employed by the health care institution. This profession already has a strong presence in many Western European countries and helps determine attitudes toward spirituality in the care of people with mental illness.
2. The introduction should also explain the distinction and similarity between spirituality and religiosity. In the course of the article, the two concepts are mixed up without it being clear whether they mean the same thing or something else. The term spirituality is also not defined, so it can evoke all sorts of associations without it being clear what you authors have in mind by it. This may also be a provisional working definition.
3. It is also striking that there is relatively little reference to the current literature in the psychology of religion that deals precisely with the conceptualization of spirituality and religiosity. It is not necessary that you repeat the discussion, but it would be important to refer to it, and at least to list what ideas have become commonplace. As a result, of course, it might also turn out - in the discussion later on - that the situation in Spain/Portugal differs from those ideas (or has some similarities).
4. At the end of the introduction, both the research question and the objective should be more clearly defined: what exactly will you investigate, and what is its purpose?
5. Line 69: What exactly does "interfere" mean here?
6. Line 80: cf. also: Churchill, S. D. (2022). Essentials of Existential Phenomenological Research. American Psychological Association. https://doi.org/10.1037/0000257-000
7. Lines 81-83: why was the study conducted in two phases?
8. Lines 85-87: what is the rationale for including respondents in both Spain and Portugal? Also in the introduction, it is not clear whether the appreciation of spirituality may be similar to that in Spain, or whether it differs. Moreover, under "materials and methods," it is not clear whether all interviews were conducted in Spanish, or whether interviews in Portugal were conducted in Portuguese. Were any Portuguese interviews translated, especially since the authors are all working in Spain?
9. Rule 87: Is it indeed a "convenience sample" or is it a "purposive sample" after all (see e.g. Churchill, p. 35)?
10. Rule 91: What does "free" mean in "free and informed consent"?
11. Line 100: is it actually a 'narrative discourse analysis' (by the way, a reference to a methodological article/handbook is missing here)? What you are doing in the study can rather be characterized as thematic analysis (cf., for example, the publications of Virginia Braun and Victoria Clarke). And if it is indeed thematic analysis, is it a deductive or rather an inductive approach?
12. Line 106: Do you mean "validation" or "triangulation"?
13. Line 109: some words are missing in this sentence.
14. Rule 117: reference to the mentioned law.
15. Line 118 ff.: it would be good to insert a table with the basic data of the respondents: gender, age, occupation, years of experience, origin, religious background.
16. Line 125: What are "counsellors? Are they spiritual caregivers or chaplains?
17. Line 129: what is meant by the "principle of circularity"? Brief explanation please, and add a reference.
18. Lines 133 ff.: Too many examples are cited per section. It seems better to limit this to two or three, choosing just the most telling quotes. As a reader, you now lose the overview.
19. Line 134: "most of them": how many respondents are we talking about?
20. Lines 134-135: Most respondents have never heard of the term spirituality, you write, and yet they associate various aspects with it. That seems like a contradiction to me.
21. Section 3.1: It is notable that none of the respondents makes a connection to religiosity.
22. Line 187: Does this quote fit the characterization above it?
23. Line 210: should be italic.
24. Rule 221: Who are these "facilitators"?
25. Line 279: "worked?
26. Line 287: ") can be omitted.
27. Lines 299-301: this contradicts lines 134-135.
28. Line 306: "they": who are we talking about here?
29. Lines 317-319: do not fit in under the heading.
30. Lines 335-337: this is a sample that is also extremely small for Portugal. What can be said here about representativeness?
31. Lines 343 ff.: Is it right to compare with literature dealing with the role of spirituality and religion in cancer? Is the role of spirituality in psychological problems and that in physical problems identical?
32. Lines 355 ff.: Is it possible to compare the role of religion and that of spirituality? See also above at point 2.
33. Rules 365 ff.: What do these results mean for your own research? After all, no such distinction was made among the respondents.
34. Lines 383 ff.: Is the argument of lack of time a correct one? Could not the question be asked here, whether this does not precisely mean that "spirituality" is not integrated into the therapeutic outlook of health care providers. If that were the case, then spiritual needs would be seen immediately and no extra time would be needed. However, the next question would be whether caregivers in general are the appropriate people to discuss those spiritual needs, and whether there should not be a separate professional to whom one can refer.
35. Rules 403 ff.: What about "chaplains" or "spiritual caregivers" in Spain and Portugal?
36. Line 410: From which course are these students?
37. Rules 426 ff.: I think that is not a conclusion from the study itself, but from the literature cited.
38. Lines 430-431: What profession do these counsellors practice? Are these "chaplains” of “spiritual caregivers”? It is not clear if this conclusion comes from the survey. It is not listed in the "results”.
39. More could be said at the end of the discussion about the limitations of the study and recommendations for follow-up research.
40. Line 448: That this study is a follow-up to a previous study also belongs in the introduction.
41. Line 457: "predictor" of what?
42. Lines 461-464: this seems to me to be a very important statement. So here is also the importance of awareness, and how best to achieve it.
Author Response
We are grateful for the time you have spent making your contributions to improve our document; they have been very thorough and interesting so that we can express our research with greater definition.
Below we respond to each of your comments:
1. We proceed to briefly clarify what you comment, specifying in which specific environment the research has been carried out (lines 77 to 86).
2. We have detailed this aspect that you comment on with the complexity of the definition of the term spirituality by providing studies that already comment on this fact (lines 52-58). Research focused on this matter that we have reviewed does not determine a clear difference between religion and spirituality, one of the issues that we consider makes them difficult, and that we consider is an objective for another type of study. We tried at all times during the writing of the article to differentiate them, but for some of the participants it means the same thing, so it is a fact that is presented as we have found it.
3. As we have mentioned, we have referred to the complexity of defining the concept of spirituality/religiosity with documents provided from different approaches (references 4-13, lines 52 to 58)
4. Added lines 93 and 94.
5. This is one of the conclusions of a previous research that we reviewed and that is available for reading and that we invite you to read if you consider it appropriate and interesting: by Diego-Cordero R, López-Tarrida ÁC, Linero-Narváez C, Galán González-Serna JM. "More Spiritual Health Professionals Provide Different Care": A Qualitative Study in the Field of Mental Health. Healthcare (Basel). 2023;11(3):303. Published 2023 Jan 19. doi:10.3390/healthcare11030303. It specifies: Other difficulties identified were the fear of not knowing how to control the situation (P-12, woman, 45 years old, nurse: “I see it as a taboo area, which is respected but better to leave it... unless they address it but without me entering”), the existing stigma when addressing the subject, (P-8, man, 45 years old, nurse: “There is a lot of stigma, even among health professionals, mental health are not given the same treatment or patients, with their mental illness credible). weighing more heavily than other aspects of their health”), the difficulty in identifying the need to address this dimension (P-10, man, 50 years old, nurse: “when a problem of these characteristics is presented to you, look the other way... and this may be present in some professionals in the sector because they are issues that perhaps make us uncomfortable and we prefer to say well.... this for another professional in another field and I'll forget about it”), or the lack of professional experience (P-4, man, 50 years old, teacher: “the truth is that you find inexperienced people who come to work in the centers who have not yet been ingrained in the concept that a person’s well-being lies in physical, psychological, social and spiritual development”).
6. We add your contribution that enriches the data, and we reference it, thank you (reference 25)
7. It was a matter of professional availability of the interviewers and the participants
8. The interviews (not written surveys) were conducted in Spanish, taking into account that both languages do not present excessive difficulty in understanding by both parties. Even so, in the moments when there were doubts, the interviewers made sure that the same understanding of the questions asked and the answers given was reached.
9. It is a convenience sample: the researcher himself chooses the subjects of the sample based on criteria of ease of access to the individuals, without including randomness in the process.
10. Written formula used to indicate that there is no coercion for participation or in the responses given
11. We have modified the paragraph for better understanding
12. Triangulation
13. No word is missing in the sentence
14. Added to the text
15. We have included more descriptive information in the text: average and age range, years of experience and range of time of experience in mental health. Sex, occupations, origin and religious affiliation already included.
16. Explained in the introduction (lines 80-82)
17. We have modified this paragraph for better understanding
18. We believe that we should keep the quotes as they are because they reflect the experience of the participants even if they are exhaustive. We believe it is relevant to show their perspectives. We have made changes to the presentation structure for better understanding.
19. We have added the number of people who had heard the term (2 people).
20. Indeed, but since it is an interview, they are invited to reflect in order to give an answer and provide more information, always avoiding biases or influences on the part of the interviewer that would spoil the objective results
21. We introduce it in the discussion (lines 377-384)
22. No, it is the specified one
23. It is already in italics
24. It refers to facilitator for spiritual care, it is clarified in the text. It is in fact one of the categories (point 3.2)
25. It is a literal phrase from the participant taken from the interview
26. Corrected
27. It is better specified to be more precise. In the line you refer to, it talks about training in mental health, not in spiritual care (lines 330-332).
28. It is specified in the text, in bold, that it refers, according to the context, to future professionals (lines 337-339)
29. It is a matter of editing. Corrected.
30. Remember that it is a sample for convenience and subject to the availability of the participants
31. It expresses interesting questions for other research. It would be interesting to establish comparisons according to health areas for future research in a more exhaustive manner
32. Same answer as in point 31
33. In fact, this category has not been made, we have believed the classification made to be more relevant from our perspective.
34. An assessment with which we agree, but it was not stated in the responses of the participants.
35. It has been referred to in the introduction
36. To do so, we invite you to read the open article: by Diego Cordero, R.; Lucchetti, G.; Fernández-Vazquez, A.; Badanta-Romero, B. Opinions, Knowledge and Attitudes Concerning “Spirituality, Religiosity and Health” Among Health Graduates in a Spanish University. J Relig Health 2019, 58 (5), 1592–1604. https://doi.org/10.1007/s10943-019-00780-3.
37. We understand your confusion, although the results in this case have been consistent with the literature.
38. It has been specified above as we have referred to
39. The limitations and prospective have been detailed in the conclusions section, but due to limitation in the extension it has been decided to point out the most relevant
40. This statement is moved to the introduction.
41. The sentence is completed
42. Thank you very much for your comment
Reviewer 4 Report
Comments and Suggestions for Authors
This article presents a qualitative study regarding an underexamined topic. In the Introduction a more detailed review of the existing relevant literature can improve the rationale of the study (e.g. there is an extended literature on older patients suffering from different mental disorders in Greece which is similar to this sample in the Mediterranean region https://scholar.google.com/scholar?hl=el&as_sdt=0%2C5&q=greece+older+adults+spirituality&btnG=). Please discuss additionally, how ethics in healthcare provision are influenced attitudes towards religion and spirituality in healthcare professionals as a relevant qualitative study has shown (https://www.igi-global.com/chapter/business-ethics-in-healthcare/335699). The interview schedule must be provided. Furthermore, methodologically it is not clear what type of analysis was followed (IPA, grounded theory, other approach)? Please clarify and describe why this type was chosen instead of the others or instead of mixed methods in the Methods section. Comparisons with other countries can make the Discussion section more interesting. Finally, tables with the main themes could be useful as well.
One last point, authors need to discuss the applications of their findings in future interventions-trainings for healthcare professionals (e.g. https://journals.sagepub.com/doi/abs/10.1177/1542305015572955 and https://www.sciencedirect.com/science/article/pii/S0885392421002372).
Author Response
Thank you for your comments so that our article better expresses our results and they are better contrasted.
- Regarding the bibliography in the introduction, which indicates that we should cite other similar studies in the field of mental health, we add others (references 24 and 25)
- In our approach, which exposes a found reality, we have not addressed how ethical aspects influence the attention to the spiritual in healthcare practice, although we believe it is very relevant and of such depth that it can be the subject of a single study, not in this case.
- Regarding the methods: A phenomenological approach was selected in order to explore the views, opinions and behaviors of mental health professionals regarding the care and attention to the spiritual needs of patients and their families in the mental health field. This approach is suitable because it enables researchers to gain insights into complex phenomena by exploring the participants' perspectives and the meanings they attach to their lived experiences. ( Sandelowski M, Barroso J. Writing the proposal for a qualitative research methodology project. Qual Health Res. 2003;13: 781–820. pmid:1289171) Related to the interview schedule, it has been noted that the interviews were carried out for 8 months at two different times: November 2021 to February 2022 and from January to February 2023.
- In your contribution of articles for the discussion, we have taken into account your suggestion of articles for future interventions in our discussion, and we are grateful for this suggestion.
Round 2
Reviewer 3 Report
Comments and Suggestions for Authors
Dear Authors,
The article is clearly improved. However, I still have some comments, because a number of things are still unclear or missing. Below, I will go through the various points that I still have a comment on. In doing so, I am keeping the original numbering. If a number no longer appears below, I no longer have any comments on it.
1. I believe it is correct to indicate in the title that the research took place in Spain and Portugal. The study is not a study that can be generalized, something that also applies to other studies in other countries. After line 79, it would be clarifying to add some sentences about how in Spain and Portugal the recognition and appreciation of spirituality in mental health care are, and to what extent the situation in the HOSJD corresponds or differs from it. Moreover, it is necessary to say something about the extent to which this situation in Spain and Portugal correspond or, on the contrary, differ. In each country, when it comes to spirituality, there is a different culture.
2. It is necessary to indicate the complexity in the text itself, especially since it is to be expected that respondents will not have a clear understanding of the concepts. There is much sloppiness in much of the literature in this area, that is, many studies do not realize enough that proper conceptualization is necessary. On the one hand, I refer to the various publications of Christina Puchalski who, as a physician, has urged, in interdisciplinary and international research, to reach a consensus regarding the definition of spirituality in the context of health care (e.g., Puchalski, C., Ferrell, B., Virani, R., Otis-Green, S., Baird, P., Bull, J., Khokhinov, H., Handzo, G., Nelson-Becker, H., Prince-Paul, M., Pugliese, K., & Sulmasy, D. (2009). Improving the quality of spiritual care as a dimension of palliative care. The report of the Consensus Conference. Journal of Palliative Medicine, 12(10), 885-904. https://doi.org/10.1089/jpm.2009.0142). On the other hand, the contributions of Zinnbauer and Pargament are still normative when it comes to the differentiation between spirituality and religiosity (see, e.g., Zinnbauer, B. J., & Pargament, K. I. (2002). Capturing the meanings of religiousness and spirituality. One way down from a definitional Tower of Babel. Research in the Social Scientific Study of Religion, 13, 23-54, and: Zinnbauer, B. J., & Pargament, K. I. (2005). Religiousness and spirituality. In R. F. Paloutzian & C. L. Park (Eds.), Handbook of the psychology of religion and spirituality (pp. 21-42). Guilford Press). I also refer to a recent article by Harold Koenig and Lindsay Carey who warn against the contamination between spirituality, religiosity, and well-being: Koenig, H. G., & Carey, L. B. (2024). Religion, spirituality and health research: Warning of contaminated scales. Journal of Religion and Health, 63, 3729-3743. https://doi.org/10.1007/s10943-024-02112-6. And further, I refer to the publications of Ninian Smart who has shown the many dimensions of religiosity/spirituality.
3. See above under 2.
7. It seems important to me to mention this ("availibity of interviewers and participants") in the text. Indeed, it could also be that the first interviews were analyzed in the meantime, and subsequent interviews were conducted with those insights - something not uncommon in qualitative research.
8. Your answer to this question also belongs in the article. It remains unclear (see also above under 1.) whether the context in Portugal is different from (or similar to) that in Spain when it comes to valuing spirituality in mental health care.
11. It remains unclear whether you used an inductive thematic analysis, or whether partly there was also a deductive approach (in fact, there is mention of a 'script' in line 118, suggesting that variables for the analysis were predetermined; see also lines 153f.). Line 121: 'qualitative thematic analysis'.
13. Line 129: If no words are missing, I don't understand the text. The sentence is not correct.
15. Lines 145-147: one decimal place is more than enough.
20. Lines 157-158: there remains a major discrepancy here: most of the participants have never heard of the term "spirituality," but know all kinds of meanings to it. How can anyone answer the question of what spirituality means when they have never heard of the term before. By the way, the latter also contradicts lines 150ff. and 486ff. Moreover, it is strange that participants who do not know the term "spirituality" do think that recognition of this dimension will particularly improve the quality of treatment (lines 330-332).
21. Does this mean, then, that no participant mentions religiosity? Cf. lines 346-347.
22. Line 212: A general complaint is related here to the stated theme. Specifically, the statement says nothing about the theme (the lack of training).
23. Line 236: should be italic.
24. Line 248: Are these "facilitators" the "counsellors"? If so, replace the word.
30. Rules 370vv.: It should be added here that these are employees of the HOSJD. It remains unclear whether there are differences between the Spanish and Portuguese branches. Noting also that the number of Portuguese respondents is very small.
In addition, some comments that are new as a result of the second version:
a. The numbering of the literature references is incorrect from 24 and 25 (both of which appear twice) onwards
b. Lines 381-383: This also very much depends on the definition of both concepts. See also point 2 above. Moreover, certainly one participant refers to church/religion (identification of spirituality with abuse in the church).
c. Line 386: 'showed' is not a correct word here. This should be: 'were of the opinion that ...'.
Best regards and good luck with the further completion of the article
Author Response
Thank you again for your comments. Below we respond to each of your comments (the changes in blue have been introduced in the text):
1. We did not add Spain and Portugal to the title, since the sample of both countries is not comparable between them due to the number of participants (40 from Spain and 5 from Portugal), which we believe would not be rigorous. Although we are aware that the clinical and sociocultural contexts are similar and especially in relation to the context where the research was developed, which are the HOSJD centres. The considerations you have referred to are better specified in the text.
2. We find the articles you cite very interesting, and we add those that we found most relevant and that provide greater solidity to what we have expressed (references 4, 5, 8, 9 and 14): the difficulty of defining the concept, something that we believe would be the objective of a different investigation, so we believe that the resulting discussion would be extensive.
3. See section 2.
7. Added.
8. We have tried to clarify this issue as best as possible in the text (lines 80 to 91)
11. Changes are made in the methods section.
13. Sorry, but we do not know which line you are referring to in the revised and corrected document. The current line 133 has been deleted.
15. This information is taken into account and a single decimal is left.
20. It is explained in the same paragraph: they do not know the term except for two people, but when asked what they understand from their position what it is, they give an answer from their own perceptions and perspectives. Throughout the interview and when addressing different questions on the subject, they are asked again whether, from those points of view, they believe that caring for the spiritual dimension in mental health is beneficial, and this is what has been reflected in the document, their opinions as they are. From our own interpretation, it may be that throughout the interview you question this aspect of care, recall your own experiences and observe in them that it is positive.
21. Yes, they name it, in the line that you yourself specify (lines 350 and following in the corrected text), and therefore, it gives consistency to previous research on the complexity of differentiating spirituality and religiosity, and the confusion that exists.
22. It is better specified in the text (in parentheses, since out of context it may not be understood), since it refers to how to identify the spiritual dimension as something complex, derived from the lack of training
23. Corrected
24. No, the participant expresses that he considers that in mental health, the physical/biological is not so essential and in therapeutic interviews it may be more accessible to talk about spirituality
30. It is described with more precision so that it is understandable. We know this institution.
a. Corrected
b. We eliminated this error.
c. The word is reveal (line 387 of the corrected document)

Reviewer 4 Report
Comments and Suggestions for Authors
Most points raised by the reviewers have been partly answered. Authors should try to provide a more detailed review of the literature in the Introduction so they can support their rationale. For example, authors can find references about patients and caregivers from similar cultural settings in
https://www.scirp.org/journal/paperinformation?paperid=98467
https://pmc.ncbi.nlm.nih.gov/articles/PMC7270639/
https://www.mdpi.com/1660-4601/17/23/9096
https://www.frontiersin.org/journals/psychology/articles/10.3389/fpsyg.2021.756080/full
https://link.springer.com/chapter/10.1007/978-3-030-32637-1_8
https://link.springer.com/chapter/10.1007/978-3-031-31986-0_35
Author Response
Once again, thank you for your contributions to help make our article a success. We wanted to clarify that the objective of our research is to understand the perspectives of health professionals regarding what they think and believe about spirituality, and the influence of spiritual care in the clinical environment where they play their role (mental health context), not how it affects them, or in relation to their own spirituality or their own mental health.
From those you suggested, we have added those that seemed most relevant to our study objective, which is spirituality and mental health and their convergence (references 4 and 17). Quality of life and aging were excluded because we believe they are other areas of analysis not related to the context of our research (changes have been made in blue in the text).
